# Generation of Negative Air Ions by Use of Piezoelectric Cold Plasma Generator

**Dariusz Korzec *** 🄳, **Daniel Neuwirth and Stefan Nettesheim**

Relyon Plasma GmbH, Osterhofener Straße 6, 93055 Regensburg, Germany; daniel.neuwirth@gmx.de (D.N.); s.nettesheim@relyon-plasma.com (S.N.)
* Correspondence: d.korzec@relyon-plasma.com

**Abstract:** The negative air ions (NAI) are used for the removal of particles or droplets from the air. In this study, three types of piezoelectric cold plasma generators (PCPG), in combination with cylindrical electrostatic ion filters, are applied for NAI production. The high voltage on the filter cylinder is induced by the electric field from the piezoelectric transformer of the PCPG. To achieve the dc bias, the cylinder of the electrostatic filter is connected to the ground over ultrafast switching diodes. The ion concentrations are measured for different airflows, PCPG powers, and electrostatic filter geometries. The NAI concentration in the order of magnitude of $10^7$ cm$^{-3}$, and a negative-to-positive ion concentration ratio of over 200 is reached. The production of ozone is evaluated and the PCPG configuration with a minimum ozone production rate is proposed. The ozone concentration below 60 ppb is reached in the airflow of 90 m$^3$/h.

**Keywords:** negative air ions; atmospheric pressure plasma; piezoelectric direct discharge; ozone



## 1. Introduction

The negative air ions (NAI) are immanently present in ambient air. The natural NAI sources include [1] radiant or cosmic rays, sunlight ultraviolet, natural electric discharges including lightning and friction-induced discharging (tribodischarging) [2], the shearing forces of water (Lenard effect) [3], and plant-based sources of ions [4].

The origin of NAIs are the negatively charged oxygen atoms O$^-$, generated mainly by electron dissociative attachment to the oxygen molecule. Once produced, they can contribute to the creation of secondary oxygen ions such as O$_2^-$ and O$_3^-$. The presence of nitrogen in the air allows for reactions ending in ions containing ionic radicals such as NO$_2^-$ and NO$_3^-$. The chemical reactions with water of air humidity allow for negative ions containing OH$^-$ ionic radicals. The presence of carbon dioxide in air results in ions containing CO$_3^-$, CO$_4^-$, C$_2$OH$^-$, C$_3$H$_4$O$^-$, HCO$_3^-$. All these ionic radicals can constitute cores of the ionic clusters containing a large number of H$_2$O molecules, as described in detail in review [5].

It is known that the NAI charges up the airborne particulate matter and allows for its electrostatic control [1]. The same mechanism is valid for aerosols [6]. This effect is utilized in numerous technical applications of the NAI, such as the charging of aerosol particles in smoke detector [7], improvement in indoor air quality by a residential air cleaner based on NAI generation [8], the removal of particles from the clean-room air [9], or neutralization of different types of smoke [10,11].

Several researchers are emphasizing the high importance of NAIs concerning the inactivation of bacteria [12]. Negative ion generators also reduce the airborne transmission of viruses [13]. Hagbom et al. [14] used an ionizer to generatie an NAI that attaches to the airborne particles or aerosol droplets. The latter, whcih are negatively charged, are electrostatically attracted to a positively charged collector plate. By using such an instrument, they achieved the effective prevention of the airborne transmission of the influenza A virus infection between animals, and the inactivation of the virus (>97%).

In many cases, the human-made instruments for NAI production imitate the natural mechanisms. Instruments for NAI production based on shearing forces of water [15], plant-based NAI release system under the pulsed electric field stimulation [1], or artificial-radioactivity-based $^{241}$Am charger [16] are known. Special paints are used to increase NAI concentration [17] or to generate negative oxygen ions [18]. Negative ions can originate from different types of electrical discharges. Examples include the corona discharge [19], dielectric barrier discharge (DBD) [20], or surface discharge microplasma [21,22].

Additionally, the piezoelectric direct discharge (PDD) can be applied to NAI generation. The PDD produces chemically active species and ions of both polarities. The electrostatic filtering of ions is used for the separation of negative ions from positive ions. Typically, an additional high-voltage supply is used to polarize the electrostatic ion filter electrodes. In this study, the filter electrode is biased by the strong 80 kHz electric field of the PCPG. It is shown that, up to the distance of 3 cm, the electric field produced by the PCPG is sufficient to bias a cylindrical floating electrode. To achieve a strong negative polarization of the cation-collecting filter electrode, the high-voltage, high-speed diodes are applied. The selective production of negative ions with concentrations exceeding $10^7$cm$^{-3}$ is demonstrated. Three different configurations of the PCPG are investigated. All experiments in this study are conducted using of the CeraPlas™ HF [23,24].

Similar to the conventional air purifiers [25,26] also PCPG show an increased level of ozone concentration. To reduce the amount of ozone produced by the PCPG, the electrode configuration similar to this used for ionic wind generation in [27] is evaluated. The PCPG configuration and process parameters are proposed, allowing the ozone concentration to remain below 60 ppb.

## 2. Experimental Details

### 2.1. The PCPG Operation Principle

The PCPG produces cold-atmospheric plasma by generating a sinusoidal signal with a frequency of 80 kHz and voltage amplitude of more than 10 kV. The electric field at the PCPG tip causes an ionization processes in the surrounding gas and the initiation of a micro-discharge [28]. The voltage transformation is achieved at resonant frequency due to the formation of a standing acoustic wave which transforms the low voltage from the input side to a high voltage at the mechanically coupled output side of the piezoelectric transformer. The operation principle of the PCPG is described in detail in the review article about the PDD [29] in the Special Issue.

### 2.2. Setup for NAI Measurement

Figure 1a shows the setup for the generation, transfer, filtering, and concentration measurement of NAIs. For plasma generation, three different PCPG configurations, presented in Figure 1b, are used and evaluated. In the first one, the standard CeraPlas™ HF package [23] is applied, powered by the CeraPlas™ drive [24]. The second one is based on a CeraPlas™ HF, partially potted in silicone, with an open CeraPlas™ tip, to allow for the generation of the PDD. The third one is analog to the second one, but with a needle-electrode glued on the tip of the CeraPlas™ HF potted using potting compound DowCorning 4241. The AlphaLab Inc., South Salt Lake, UT 84115, USA, air ion counter is used to measure the negative and positive ion concentrations. The distance between the tip of the CeraPlas™ HF element and the windshield of the ion counter in the configuration, as shown in the picture is 15 cm. This distance is a compromise between the influence of the electric field of the PCPG on the air ion counter, when the distance is shorter, and the strong decay of the ions, if the distance is larger. As will be shown in Section 3.1.1, this optimum depends on the investigated range of air velocity. The PCPG is positioned in a PMMA tube with an inner diameter of 100 mm. Airflow in the PMMA tube is established using the ebmpapst 4414M fan, whose air flow rate is typically 100 m$^3$/h = 1670 NLM. The actual air velocity was measured using the air velocity transmitter EE575-V3C2 positioned

at the axis of the PMMA tube. The air velocity $v_{air}$ was controlled by changing the voltage of the fan $V_{fan}$ from 18 to 28 V, accordingly to the fitting line given as:

$$v_{air}[m/s] = 0.1812 \cdot V_{fan}[V] + 1.328. \tag{1}$$

To remove the positive ions from the airflow, an exchangable electrostatic ion filter, in the shape of an aluminum cylinder, is applied, with a diameter of 32 mm and a length varying from 2 cm to 8 cm. The default value for length is 40 mm. In a standard setup for ion filtering this cylinder would be biased, with negative potential from the external voltage supply to promote the drift of positive ions toward the cylinder. In the setup presented here, the characteristic property of the piezoelectric transformer is used to simplify the ion source design. The high voltage tip of the piezoelectric transformer generates a strong alternated electric field, which is measurable in the distance of many centimeters from its source [30]. Consequently, at the surface of the metal cylinder positioned in the vicinity of the HV tip of the piezoelectric transformer the potential oscillation is induced. Connecting the metal cylinder over a diode to the ground results in a dc bias, which is one of the points investigated in this study. The cylinder is grounded over number $n$ of serially connected diodes MUR 1100E-GA [31] with 900 V reverse voltage, oriented with cathodes to the ground, as shown in Figure 1a. The voltage measured on the filter cylinder by use of the voltage probe Tektronix P6015A with an input resistance of 100 MΩ and input capacitance of less than 3 pF, is a sinusoidal oscillation with the frequency 80 kHz, between $-100$ V and $-600$ V. Consequently, for these conditions, the filter average voltage is $-350$ V. Unfortunately, due to a comparable input impedance of the reverse polarized diodes (an input resistance of 100 MΩ and the input capacitance of less than 2 pF), much higher actual voltage values established on the cylinder than the measured one can be expected, and are not included in this paper.

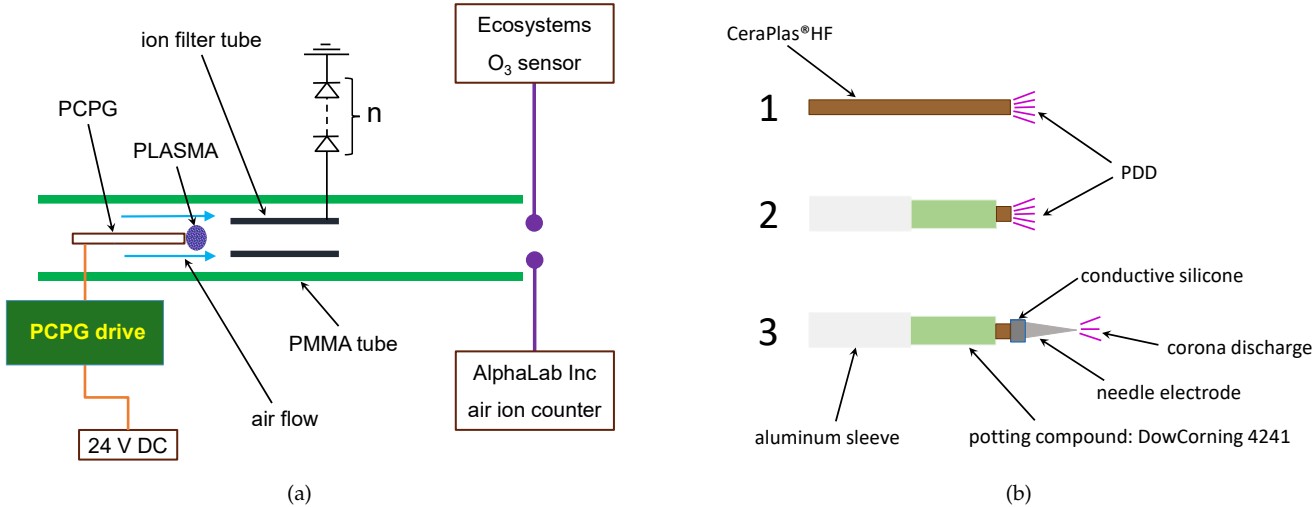

(a)                                                                                    (b)

**Figure 1.** Setup for measurement of ion and ozone concentration. (**a**) Geometry of the measurement system. (**b**) Three configurations of piezoelectric cold plasma generator (PCPG) 1 non-potted CeraPlas™ HF, 2 potted CeraPlas™ HF, and 3 CeraPlas™ HF driven corona discharge.

### 2.3. Ozone Concentration Measurement

Due to its high accuracy in a broad range of ozone concentrations, UV absorption spectroscopy is frequently used [32,33]. For the ozone concentration measurements presented in this work, the ozone Analyzer Model UV-100 of ECO Sensors, Inc., Newark, CA, USA is used, allowing for ozone concentration measurements in the range from 0.01 to 1000 ppm (volume). It locally samples the air with a pumping speed of about 1 L/min through PTFE tubing. Such a solution allows for a precise definition of the place, in which the ozone concentration is determined.

### 3. Results and Discussion

*3.1. Non-Potted PDD*

3.1.1. Influence of Airflow

The influence of the airflow on the ion concentration was investigated without the ion filter. The results in Figure 2 show the dependence of the ion concentrations on the air velocity. The measurements are conducted for two power values of the PCPG in configuration 1, as defined in Figure 1b. As expected, the ion concentration is higher for a higher power. For both power values and both ion types, the concentration increases with increasing airflow rate. This result is not obvious, because the dilution effect can be expected, resulting in a decrease in the concentration. The observed contradictory behavior can be explained on the basis of the short dwell time of the ions. The dwell time is determined by three main loss mechanisms [34]: (i) the volume recombination of negative and positive ions, which is important for high ion concentrations, (ii) losses of ions on the wall of the PMMA tube, and (iii) loss of ions by diffusion onto aerosol particles (coagulation sink) [35].

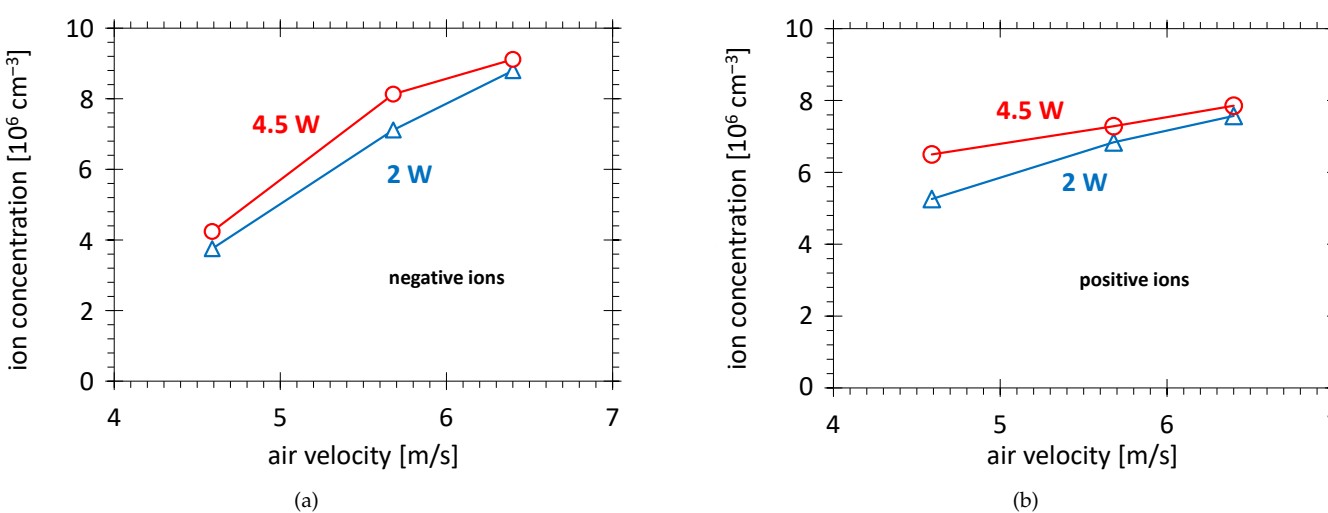

**Figure 2.** The ion concentration vs. air speed for (**a**) negative and (**b**) positive ions. The PCPG configuration 1, without filter, for two CeraPlas™ HF power values.

Let consider the timescale of the losses due to the ion–ion recombination. The simplified balance equation can be written as

$$\frac{\mathrm{d}n_{\mathrm{ion}}}{\mathrm{d}t} = -\alpha n_{\mathrm{ion}}^2 \tag{2}$$

where $\alpha$ is the ion–ion recombination coefficient, and $n_{\mathrm{ion}}$ is the ion concentration. At room temperature, $\alpha = 2.5 \times 10^{-6}$ cm$^3$s$^{-1}$ can be assumed [34]. Taking $n_{\mathrm{ion}} = 8 \times 10^6$ cm$^3$, the ion decay times in the range of few tens of ms can be obtained from Equation (2). This is comparable with the time of 25 ms, needed for airflow to cover the distance of 15 cm. With increasing air velocity, the ions need less time for transfer from the PCPG to the ion sensor. Consequently, fewer ions are lost from the air stream.

The positive ions generally behave very similarly to the negative ions (see Figure 2b). However, the increase in the concentration with air velocity is, for negative ions, about four times higher than for positive ions. The electrons possibly play no crucial role in the explanation of this effect, because their decay time is less than 10 μs [36], which is much shorter than the time of 25 ms that is needed for airflow to cover the distance of 15 cm. The proposed explanation of this effect is based on the difference in diffusivity of positive and negative ions. The NAIs tend to build up large clusters, which are heavier and less mobile than the positive ions. Consequently, their radial diffusion toward the PMMA wall is slow,

and a distinct maximum of NAI concentration at the axis of the PMMA tube can sustain longer than for positive ions, reaching a quite time-stable radial distribution in a short time. When increasing the air velocity, the maximum NAI concentration can reach the air ion counter. No such effect occurs for the more homogeneously distributed positive ions.

### 3.1.2. Influence of Number of Diodes

In the investigation of the influence of the number of diodes $n$ (see Figure 1a) on the ion concentration, the PCPG configuration 1, as defined in Figure 1b, is applied. The case of $n = 0$ can be interpreted in two ways. The cylinder can be left electrically floating or it can be grounded. For the floating cylinder, the concentrations of positive and negative ions are almost the same, with a negative-to-positive ion concentration ratio of 1.06. For the grounded cylinder, the negative-to-positive ion concentration ratio is close to the unit, reaching 0.928. The conclusion is that both floating and grounded filter cylinders showed no significant ion filtering action. However, the concentration of both ion types, in the case of the floating ion filter cylinder, is significantly larger than in the case of grounded one. It is about $10^7$ cm$^{-3}$ and half of this value, respectively. This difference can be explained by the net current flowing to the ground in each electrical half cycle of the electric field for the grounded ion filter cylinder, causing the ion losses and consequently lower ion concentrations.

With one diode conducting from the cylinder to the ground the negative ion concentration is 21 times higher than the concentration of positive ions. For two diodes, this ratio is 250, and for three diodes—100 (see Figure 3a). The optimum for two diodes is a result of two opposing tendencies. On the one hand, with an increasing number of diodes, the negative voltage on the ion filter cylinder in the negative half-cycle of the voltage on the tip of the PCPG is increasing. On the other hand, the increasing resistivity of the serially connected diodes causes an increase in the positive voltage during the positive half-cycle and, consequently, a reduced dc bias of the ion filter cylinder.

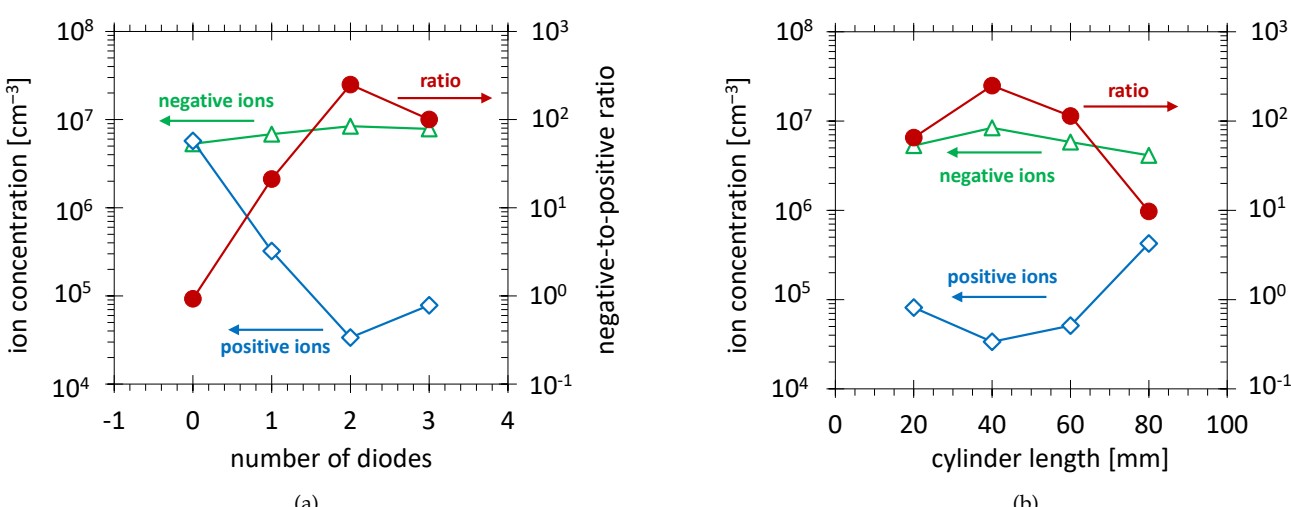

(a)                                                    (b)

**Figure 3.** The ion concentrations and negative-to-positive ion concentration ratio for (**a**) different number of diodes at ion filter cylinder length of 40 mm, and (**b**) varying length of the ion filter cylinder with two diodes. Measurements are conducted in PCPG configuration 1 with the power of 4.5 W.

### 3.1.3. Influence of Cylinder Length

To optimize the ion selectivity, the length of the ion filter cylinder is varied. For this experiment, the PCPG configuration 1, the fan voltage of 24 V, the PCPG power of 4.5 W and two diodes are applied for connection between cylinder and ground. The results are summarized in Figure 3b. This dependence shows a maximum. The ion filter cylinder with a length of 4 cm shows the highest negative ion concentration and the highest negative-to-positive ion concentration ratio. This value is also a result of two opposite tendencies. With

the increasing length of the filter cylinder, the amount of positive ions extracted from the airflow is increasing, resulting in an increase in the negative-to-positive ion concentration ratio. For excessively long cylinders, the negative potential of the ion filter cylinder is retarding the negative ions, resulting in a prolonged time-of-flight and higher losses.

*3.2. Potted PCPGs*

In this section, the performance of potted PDD ( PCPG configuration 2) and corona discharge powered by the piezoelectric transformer (PCPG configuration 3) for NAI production is examined.

3.2.1. Potted vs. Non-Potted

The potted PCPG (configuration 2 in Figure 1b) reaches, for 4.5 W, an NAI concentration of about $4.6 \times 10^6$ (see Figure 4a), which is only about half of the value $8.4 \times 10^6$ (see Figure 3) reached under comparable conditions for the non-potted PCPG (configuration 1 in Figure 1b). The reason for the lower efficiency of configuration 2 is the mechanical damping of the resonant oscillation of the piezoelectric transformer and the dielectric losses in the potting material. Despite the better performance of configuration 1, the potted PCPGs are included in this study, because they have the advantage of higher mechanical robustness.

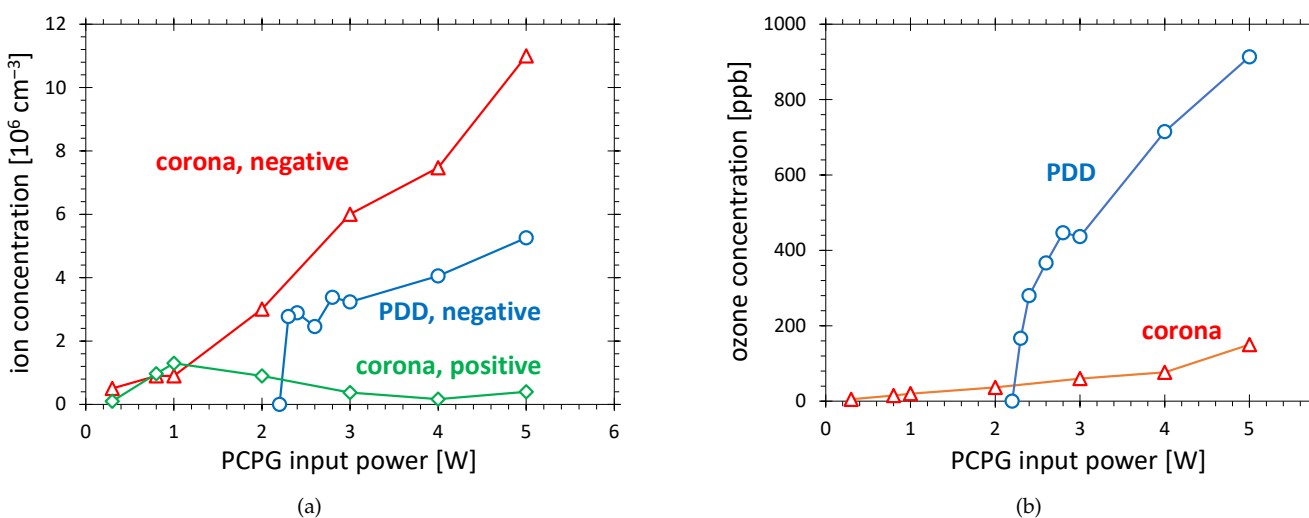

(a)                                                                 (b)

**Figure 4.** Comparison of the ion and ozone concentration for PDD (configuration 2) and corona (configuration 3) as a function of PCPG power. (**a**) Ion concentration and (**b**) ozone concentration.

3.2.2. Influence of Power

Figure 4a shows the negative ion concentration as a function of PCPG power for configuration 2 and 3. Both curves (red and blue) show the increase in the NAI concentration with power.

Configuration 3, with the corona needle, has two important advantages compared with configuration 2. First, it doubles the high values of NAI concentration that existed for configuration without the needle. For 4.5 W, it is slightly higher than for the non-potted device (configuration 1). Second, it sustains the discharge and generates NAIs for the much lower power level of 0.3 W. For configuration 2, no measurable ions are produced for power below 2.3 W.

Without the filter, both the negative and positive ion concentrations increase. If the ion filter is applied, a strong suppression of the positive ion flow can be observed (see the green line in Figure 4a). The main reason for such behavior is the increased voltage of the ion filter cylinder. The higher oscillation voltage of the PCPG tip induces oscillations with higher voltage on the ion filter cylinder, and consequently leads to higher negative DC bias

and stronger filtering action. For the power of 1 W and less, the polarization of the filter is too weak to sustain the filtering action.

### 3.2.3. Ozone Concentration Control

The production rate of ozone in configuration 1 reaches 0.1 g/h. Looking at Equation (1) in [29], it can be seen that, at the air flow of 1500 SLM (90 m$^3$/h), the ozone concentration in the air stream resulting from such production rate is 5 ppm. This is two orders of magnitude more than the value of 60 ppb that needs to be reached. The means for ozone level reduction have to be applied. The first step could be the application of configuration 2, because it generates less ozone. The blue curve in Figure 4b shows that, even for maximum power, the ozone concentration is below 1 ppm. Unfortunately, it is not possible to reduce the ozone concentration below 60 ppb by a reduction in power, because it is abruptly disappearing below 2.3 W, together with the NAI concentration (see the blue line in Figure 4a).

The characteristics of configuration 3 are much better in this respect. The corona discharge driven by the piezoelectric transformer is already present for the power of 0.3 W. For the maximum power of 5 W, it reaches an ozone concentration of 150 ppb, which is six times lower than the PDD, but about three times higher than the targeted value of 60 ppb. However, in the power range from 0.3 W to ca. 2.5 W, the ozone concentration below 50 ppb is measured and, at the same time, the considerably high negative ion concentration of $5 \times 10^6$ cm$^{-3}$ is reached.

The corona discharge can be sustained for much lower power than for PDD due to the much higher electric field on the tip of the needle electrode, than on the front surface of the PCPG.

Several aspects should be considered in the explanation of the much lower ozone production with corona than with PDD.

The first is the temperature of the plasma-generating electrode. Since, in PDD, the discharge is distributed over an area of several square mm of the PCPG, the thermal load is small and the ceramic surface is never hotter than 90 °C. The radius of the corona tip is 0.1 mm, and locally reaches the temperature up to several hundred °C. It is well known that an increased temperature causes less ozone production in the discharge [37].

The second is the power density of the discharge, which is higher in the corona discharge. At a higher power density, higher local production rates of ozone can be expected. Consequently, at the micro-scale, the ozone concentration can be reached, which causes the self-destruction of ozone and finally lowers the global ozone production rate. This is not the case in the PDD discharge, which occurs in comparatively larger volumes, and the ozone concentrations promoting self-destruction are not reached.

The third is the orientation of the micro-discharges concerning the airflow. In corona, the discharge develops in the direction parallel to the airflow resulting in a long time needed to remove the ions out of the discharge. This time is much shorter in the PDD because the micro-discharges develop almost perpendicular to the airflow direction. Also, the cooling effect of the airflow is in the PDD stronger.

## 4. Conclusions

The piezoelectric cold plasma generators (PCPGs) are promising devices for generation of negative air ions.

The PCPGs generate positive and negative ions, which are transferred to the air flow and are reaching ion concentrations in the range of $10^7$ cm$^{-3}$. The concentrations increase with increasing air flow and increasing power.

A negatively biased metal cylinder can change the balance between the ion types, acting as the electrostatic ion filter. The negative bias can be induced by the electric field of the PCPG if the metal cylinder is connected to the ground over diodes, with diodes oriented with the cathode to the ground. A negative-to-positive ion concentration ratio of 250 is demonstrated. This maximum is achieved for two diodes with 900 V reverse voltage,

connected in series. The optimum filtering effect was achieved for a metal cylinder with length of 40 mm.

To improve the mechanical robustness of the system, the PCPGs based on potted CeraPlas™ HF devices are investigated. The potted version of the PCPG reaches the factor of two lower NAI concentrations than the non-potted version. However, the potted version with a needle electrode fixed on the tip of the piezoelectric transformer, by use of the electrically conducting potting compound, reaches a higher NAI concentration than the standard configuration. Moreover, with this configuration, the ozone production rate can be achieved, allowing anozone concentration below 60 ppb to be reached in the air flow of 90 m$^3$/h.

**Author Contributions:** Conceptualization, D.K., S.N. and D.N.; methodology, D.K. and S.N.; software, D.K.; validation, D.K., D.N. and S.N.; formal analysis, D.K. and D.N.; investigation, D.K. and D.N.; resources, S.N.; data curation, D.K.; writing—original draft preparation, D.K.; writing—review and editing, D.K. and D.N.; visualization, D.K.; supervision, S.N.; project administration, D.N.; funding acquisition, S.N. All authors have read and agreed to the published version of the manuscript.

**Funding:** This research received no external funding.

**Institutional Review Board Statement:** Not applicable.

**Informed Consent Statement:** Not applicable.

**Data Availability Statement:** Data supporting reported results can be obtained on request from the corresponding author.

**Acknowledgments:** The CeraPlas™ HF devices and CeraPlas™ drives are provided by TDK Electronics GmbH.

**Conflicts of Interest:** The authors declare no conflict of interest.

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
