# Peer review of "Generation of Negative Air Ions by Use of Piezoelectric Cold Plasma Generator"

_plasma, doi:10.3390/plasma4030029_

Round 1
Reviewer 1 Report
Dear authors,
After reviewing your manuscript, I would like to state that you managed to prepare a relatively comprehensive view of the possibility of generating NAI using a plasma generator of different configurations based on a piezoelectric phenomenon. You also detect and determine the concentration of NAI and the ratio of negative-to-positive ions and measure ozone concentration. On the other side, I see the manuscript as a relatively brief report of your results without in-depth discussion. I have the following suggestions for editing and comments on the manuscript:
- all authors belong to the same affiliation, "Relyon Plasma GmbH"; therefore, the row with the "2nd" affiliation is unnecessary and should be deleted
- row 11: the semicolon at the end of keywords is unnecessary
- row 45: typo in the word "form", it must be corrected to the "from"
- the description of the Figure 1: I recommend to extend it by the adding the listing of the configurations 1-3
- row 77: "...generating the 80 kHz signal with high voltage" - this wording seems inaccurate to me - "signal with high voltage?", maybe should be reworded
- row 78: 10 kV - What does this value mean? The amplitude or peak-to-peak voltage value?
- row 82: "...in detail in this special issue in the review article", maybe should be reworded to "...in detail in the review article [29] of the special issue"
- row 93: Could the authors explain and eventually add this information also to the manuscript why they decided for 15 cm as the distance between the tip of the CeraPlasTM HF element and the windshield of the ion counter in the investigated configuration?
- Is this distance 15 cm optimal also in the case of higher flow rates of air?
- Figure 2: It would be beneficial to point out why "fan voltage" in the range" 18-28 V is investigated in the graphs.
- rows 124-126: Could the authors explain why the possitive ions have the longer dwell time in the comparison to the negative ions?
- row 129: "Figure 1a)" instead of "1b)" should be referred here, in my opinion.
- row 190: What is the meaning of the abbreviation "PT"?
Finally I would like to state that after minor revision I recommend the manuscript accept for publication in the "Plasma" journal.
Author Response
see the magenta comments in the uploaded Word document

Reviewer 2 Report
The manuscript aims at characterizing a piezoelectric transformer as a source for negaticve ions.
It describes the use of a commercial piezo transformer as atmospheric ion source, which is somehow relevant being these sources of pratical interest in the generation of atmospheric cold plasma. Nonetheless, the manuscript features some weakness in the experiment describtion and in the interpretation of the results.
I would recommend it for pubblication after some revisions.
Comments follows hereafter.
2.2 (and fig 1). It is not clear the arrangement of the “ion filter”. Line 98, “reverse voltage” maybe refer to “breakdown voltage”. One can imagine that adding diodes in series will rise the (negative ) potential value of the filter with respect to ground, but this is not discussed, neither is discussed the charging mechanism (capture of negative ions? Electron drift?). The point is briefly discussed in lines 135-145 but the authors, if possible, should add some more informations.
Another weak point is the effectiveness of putting diode in series without a balancing network, and the fact that the potential measured never reaches the breakdown voltage of a single diode (I guess this is due to the finite value of the probe impedance, but still the point is not explained).
Figure 2. Is there a way to express the abscissa in terms of air speed, flow or something similar? I got there’s a proportionality but the use of fan voltage really does not read well. Anemometric probes are quite cheap…
3.1.1 In the discussion of ion concentration increase (lines 119-122) some number on negative ion lifetime would refinforce a discussion which is reasonable but purely qualitative.
Figure 3 Please consider adding a legend to improve readability.
3.2.3 The behaviour of the system in corona mode is quite interesting , also in conideration that normal operation probably would generate ionization also by field effect, but on the crystal sharp edges rather than in a single point (so maybe there’s a sort of volume effect)
The manuscript would benefit from some interpretation of the lower production of Ozone. Can you eventually show any data on the UV emission, or comment on the role of free electrons?
Author Response
See the comments in magenta in the uploaded Word document.
